# Comparison of Self-Etching Ceramic Primer and Conventional Silanization to Bond Strength in Cementation of Fiber Reinforced Composite Post

**DOI:** 10.3390/ma12101585

**Published:** 2019-05-15

**Authors:** Nan-Sim Pang, Bock-Young Jung, Byoung-Duck Roh, Yooseok Shin

**Affiliations:** 1Department of Advanced General Dentistry, College of Dentistry, Yonsei University, Seoul 03722, Korea; pangns@yuhs.ac (N.-S.P.); jby1004@yuhs.ac (B.-Y.J.); 2Department of Conservative Dentistry, College of Dentistry, Yonsei University, Seoul 03722, Korea; operatys16@yuhs.ac; 3Department of Conservative Dentistry, Oral Science Research Center and Microscope center, College of Dentistry, Yonsei University, Seoul 03722, Korea

**Keywords:** Self-etching ceramic primer, FRC post, push-out bond strength, surface treatment

## Abstract

Various mechanical and chemical surface treatments have been proposed to improve the retention of fiber-reinforced composite post (FRCP), but the results are still controversial. The bond strength and durability of a self-etching ceramic primer, which was recently released as an alternative to etching and silane, are not yet known. This study aimed to compare and evaluate the push-out bond strength of different surface treatments of FRCPs after an artificial aging procedure. Four groups (n = 10) were established to evaluated FRCP surface treatments (dentin adhesive bonding; silane and adhesive bonding; hydrofluoric acid, silane and adhesive bonding; and a self-etching ceramic primer). They were bonded with dual-curing rein cement (Multilink N) and stored in distilled water at 37 °C for 30 days, then thermal cycled for 7500 cycles. After being sectioned into 1 mm thickness, each coronal and apical part was evaluated for its the push-out bond strength by a universal testing machine. Each debonded specimen was observed by an optical microscope and divided according to the failure modes. The results showed that silane treatment significantly improved push-out bond strength, but the self-etching ceramic primer did not do so. Additional hydrofluoric acid treatment or the adhesive bonding agent alone did not significantly improve the retention of FRCPs. Cohesive failure of the luting material was found most frequently in all groups.

## 1. Introduction

Prefabricated posts were introduced in the 1950s [1,2]. The initial products were made of metal, mainly stainless steel, which was followed by titanium alloys. Metals have been chosen because of their inherent mechanical properties, with the aim of increasing the strength of the remaining dental structure. However, metal posts tend to be less aesthetic and corrosive (unless they are noble alloys) and have the disadvantage of impairing the appearance of final aesthetic restoration [1,3]. Above all, subsequent studies have revealed that stiff metal posts could transmit the stress load directly to the roots and cause root fracture [3,4]. This problem resulted from a mismatch of the modulus of elasticity between a stainless steel post (200 GPa) and dentin (20 GPa). Clinically, this combination of two materials with different moduli of elasticity often caused occlusal stress to concentrate between their interface, resulting in irreversible vertical root fractures [3,4]. Ideal posts should have mechanical properties and a behavior such as that of the tooth structure. 

Fiber reinforced composite posts (FRCPs) were introduced in the early 1990s as an alternative to cast posts and metal dowels to restore endodontically treated teeth [3]. Fiber posts are recommended to adhere to the intracanal dentin by an etchant, primer, adhesive, and resin composite technique. The FRCP can form a homogenous structure in the dentin wall with resin cement and other resinous restorative materials [5]. The post-core tooth monoblock, which would theoretically induce stress reduction by the distribution of functional loads [6], would reduce the risk of root fracture [5]. Their modulus of elasticity ranges from 16 to 40 GPa [3], with a stiffness to dentin [7]. FRCP also has a translucent property, allowing it to transmit light energy along the entire length of the post space and enhance the polymerization of the photo-activated adhesive system [8]. Therefore, posts luted with dentin-bonding adhesive systems exhibited less microleakage than those cemented with non-dentin bonding cements such as zinc phosphate or glass ionomer [9].

Simultaneously, the methods of FRCP management and the selection of resin luting agents have emerged as crucial elements of the post cementation phase. This is due to frequent adhesive failures, such as a loss of retention of the post [10]. The bonding effect among dentin, resin luting agents, and posts can affect the retention of post-core restoration. FRCP is usually bonded to the intracanal dentin wall by bonding agents and resin cements with flexibilities similar to FRCP to increase retention and improve their mechanical properties in the restored teeth [11]. 

In general, the post type [12], the cement properties [13], and the cement bonding effect to the post, and root canal dentin [14] are related to the retention of FRCPs. Self-adhesive resin cements and dual polymerizing resin cement are suitable materials for bonding FRCPs to the root canal dentin wall due to their higher initial bond strength and less solubility than water-based cements. In detail, there are several factors that affect the bond strength at the post cement root interface, including the degree of hydration of root canal dentin, the use of eugenol-containing sealers, the surface conditioning agent and luting cement used, the cavity configuration factor, the orientation of the dentinal tubules, and the anatomical density at different levels of the root canal [15,16]. 

Among them, previous studies have investigated various methods using chemical and mechanical surface treatments to improve the interfacial bond strength between posts and resin cements. These methods may be divided into three categories: Mechanical treatment such as silica coating, airborne-particle abrasion, and acid etching; chemical methods, such as silanization and/or adhesive application; and treatments that combine both a micromechanical and a chemical component [17]. 

As a method for obtaining micro-mechanical interlocking between the resin cement and FRCP, airborne-particle abrasion or acid etching have been investigated for their effect on bond strength [11,18]. Micromechanical surface treatments have been shown to improve the bond strength between composite resin core materials and FRCP, in general [14]. However, it has also been reported that the application of hydrofluoric acid affects the integrity of the fiber post because it caused microcracks, discontinuities, and fractures in the fiber layer without improving the shear bond strength between the composite resin and the fiber post [16,17]. The retention of posts luted with resin cement may be more adversely affected by the chemically treated dentinal walls of the endodontic treated canal [19]. 

Chemical bonding between the resin cement and FRCP has also been studied; typically, this has been in the form of silanization. Silane coupling agents have an organic functional group that allows covalent bonds to form between the resin cement and the quartz fibers of the post, which contributes to increase the bond strength between ceramics and resin composite materials [8]. While some reports showed that silanization improved the microtensile bond strength between resin cement and FRCPs, there was no significant difference in the bond strength between silane-treated posts and untreated posts in other studies [20,21]. As for the post surface treatment methods, investigations into the efficiency of silanization for improving bond strength have reported conflicting results. 

Meanwhile, a self-etching ceramic primer (Monobond Etch & prime, Ivoclar Vivadent) as a single-component ceramic primer, has been introduced to the market, as an alternative to hydrofluoric acid etching/silane coupling agent routine treatment. This product integrates the etching and silane priming treatments in a single step. It has been shown to shorten the treatment time of the clinical steps by etching and silanating glass-ceramic surfaces in one working step. This self-etching ceramic primer significantly reduces the processing time of all ceramic materials compared to the conventional procedures without the loss of bond strength. Furthermore, the technique sensitivity or inaccuracy of the pre-treatment of glass-ceramic restorations compared with conventional conditioning is reduced. This self-etching ceramic primer contains 15% to 25% ammonium poly fluoride as a conditioning agent to etch the ceramic surface. This acidic component would be responsible for the partial dissolution of the glassy phase of the ceramic, thus a superficial etching pattern was obtained compared with conventional hydrofluoric acid (HF) processing. In spite of the less-pronounced etching pattern, the mean values of the bonding strength were not statistically significantly different from that obtained with conventional HF etching. This is because the ammonium poly fluoride ions induce the formation of reactive silanol groups. Self-etching ceramic primer is a combination of ammonium poly fluoride and trimethoxysilylpropyl methacrylate (methacrylate silane), responsible for helping the chemical adhesion. Rinsing ceramics removes polyfluoride and thus the silanol groups are no longer stabilized. This leads to a highly efficient functionalization process that offsets the less pronounced etching patterns. Both micromechanical interlocking and chemical boding ensure a bond strength similar to conventional processing [22]. 

This self-etching ceramic primer can be used with all methacrylate-based luting composites and all glass ceramics. The self-etching ceramic primer showed significantly higher shear bonding strength (SBS) values with polymer-infiltrated-ceramic network blocks, which suggests that it can be used as a primer for bonding polymerized resin materials [23]. However, it has not been sufficiently reported whether this simplified silanization can be applied to FRCPs and their long-term stability. 

The objective of this study was to evaluate the push-out bond strength and failure modes of differently treated FRCPs and to compare the effects of acid etching, silanization, and especially, the self-etching ceramic primer.

## 2. Materials and Methods

### 2.1. Specimen Preparation

Forty human mandibular premolars that had been extracted for periodontal disease were stored at 4 °C in 0.1% thymol solution (Sigma-Aldrich, St. Louis, USA) to keep the surface wet for less than 2 months. Only those premolars with single root and a single canal of vital pulp were selected for this study. Roots with distinctly oval canals were excluded from the study. 

Forty teeth were endodontically treated and then FRCPs (D.T. Light-post #2, Bisco) were subjected to different surface treatments and adhesively luted in the post spaces. The root canal length was standardized by sectioning the teeth below the cement enamel junction to a length of 14 mm. These canals were prepared using ProFile Ni-Ti rotary instruments (Dentsply Maillefer, Ballaigues, Switzerland) with 06 taper up to #40 size. They were irrigated during endodontic treatment with 5.25% sodium hypochlorite. After canal enlargement and shaping, the canals were obturated with gutta-percha and a resin-based sealer (AH plus; Dentsply Sirona) using a continuous wave compaction technique. Subsequently, the access cavity was sealed with modeling wax (Henry Schein) and the roots were stored in 0.1% thymol solution at 36 °C for at least 72 h. 

### 2.2. FRCP Cementation Procedure

Gates-Glidden burs (Dentsply Maillefer) were used to remove 10 mm of gutta-percha from the root canals. The post space was then prepared using a green-colored LuxaPost preparation drill with a diameter of 1.5 mm. The post spaces were irrigated with a 5.25% sodium hypochlorite solution, rinsed with distilled water, and dried with paper points. The mixed Multilink Primer A/B was applied onto the dentinal canal surface using a microbrush and scrubbed in for 30 s. 

Forty FRCP posts were divided into 4 groups (n = 10) according to the surface treatment used for each group. In group 1, dentin adhesive bonding agent (One step, Bisco) was applied on the surface of the post. Thereafter, the post was dried gently. For group 2, prior to bonding adhesion, the post surface was silanized using a silane coupling agent (Bis-silane, Bisco). The silanized post was then dried at room temperature (21 °C) for 60 s from a distance of 30 cm using an air-syringe. In group 3, prior to silanization, the post surface was etched with 5% hydrofluoric acid (IPS ceramic etching gel, ivoclar vivadent) for 60 s and rinsed with water spray. Acid etching and drying were followed by silanization and adhesive bonding. For group 4, self-etching glass-ceramic primer (Monobond Etch & Prime, ivoclar vivadent) was responsible for acid etching and silanization. The self-etching ceramic primer was applied to the post surface using a microbrush, rubbed into the surface for 20 s, and left for 40 s without agitating. It was rinsed off from the surface with a water spray and dried with compressed air. 

As previously mentioned, after the surface treatment of each group, resin cement (Multilink N, ivoclar vivadent) was used for cementation of the FRCP posts. Multilink N was injected into the post space, after which an FRCP post was inserted and light-polymerized at a distance of 2 mm for 40 s at 1000 mW/cm^2^ (VALO light irradiator; Ultradent Products Inc, South Jordan, USA).

### 2.3. Artificial Aging Procedure

After post cementation, the coronary part of the exposed dentin and post were completely covered with glass ionomer cement (Fuji IX, GC corp). For artificial aging, all specimens were stored in distilled water at 37 °C for 30 days, then thermal cycled for 7500 cycles (5 °C/55 °C) with a dwell time of 30 s and a transition time of 6 s.

### 2.4. SEM Analysis

The surfaces of the specimens were analyzed by scanning electron microscopy (JSEM-820, JEOL, Tokyo, Japan). The specimens of all groups were mounted after carbon adhesive application and coated with gold palladium. Representative images were obtained from each group at 100 X and 500 X magnification.

### 2.5. Push-Out Bond Strength Evaluation

The specimens were sectioned horizontally into 1 mm thick slices by using a low-speed diamond saw (Met-Saw; R&B Co Ltd.) Each coronal and apical part was analyzed to evaluate the push-out bond strength. The push-out bond strength was tested by a universal testing machine (EZ-S; Shimadzu Scientific Instruments, Kyoto, Japan). The test was performed by directing the load from the apical to the coronal direction at a cross-head of 0.5 mm/s until bond failure occurred. The bonding force value (N) of each section was divided by post-dentin surface area to calculate the push-out bond strength value (MPa). The method of calculating the bond surface area has been described in previous studies [24,25]. A schematic flow of the entire process is shown in Figure 1.

### 2.6. Microscopic Evaluation

After the push-out bond strength test, the failure mode of each debonded specimen was analyzed by two independent operators using an optical microscope (OPMI Pico; Carl Zeiss AG, Jena, Germany). The failure modes were divided into 5 criteria [26]: Adhesive failure between dentin and luting cement; adhesive failure between luting cement and post; cohesive failure within luting cement; cohesive failure within the post; and mixed failure. Additionally, scanning electron microscope images were obtained from the impression of the specimens.

### 2.7. Statistical Analysis

The debonding force and the push-out data were first verified using the Shapiro-Wilk test for normality of data distribution and by Levene’s test for the homogeneity of variances. They were compared with a two-way analysis of variance with post treatment (group) and location (coronal versus apical portion), followed by the Tukey Honestly Significant Difference (HSD) test for post-hoc analysis at α = 0.05. Failure modes were analyzed using Fisher’s exact test (α = 0.05). All statistical analyses were conducted using software (SAS v9.2; SAS Inc, Chicago, IL, USA).

## 3. Results

Figure 2 shows representative surface morphologies of all groups. At high resolution (x500 magnification), group 1 (dentin bonding) showed that irregular and superficial coating materials were attached on the post matrix embedded into fibers. The SEM of group 2 (silane + dentin bonding) showed more invasive bonding material into the post fiber and matrix. The SEM of group 3 and 4 revealed rough coating surfaces, the discontinuity of the fibers and the irregular destruction of the matrix component.

As shown in Table 1 and Table 2, the two-way ANOVA showed independent significant effects of the surface treatment of the post and the location of the post. There was no significant interaction between the surface treatment and the location of the post in both the debonding force and the push-out bond strength.

Table 3 presents the means and standard deviations of the debonding force (N) and the push-out bond strength values (MPa) achieved upon dislodging the posts from the two regions (coronal/apical) in each post surface treatment group. The self-etching glass-ceramic primer and dentin bonding groups attained lower bonding strengths than the HF and silane treated group or the silane-treated group (*p* < 0.05 Table 3). Specimens from all groups exhibited significant differences between the coronal and the apical parts (*p* < 0.05, Table 3).

Post surface treatment was a significant factor for debonding force and push-out strength. Mean values with the same alphabetic letter (A or B) were not statistically different (*p* > 0.05).

Post location was also a significant factor for debonding force and push-out strength (*p* < 0.05, different symbols (¢ and §) mean statistical significance).

DB, adhesive dentin bonding; HF, Hydrofluoric acid.

Data presented as n (%). Fracture mode: 1, adhesive failure between the luting material and post; 2, adhesive failure between luting material and dentin; 3, cohesive failure of fiber-reinforced composite post (FRCP); 4, cohesive failure of luting material; 5, mixed type fracture. DB, adhesive dentin bonding; HF, Hydrofluoric acid.

The fracture modes of the specimens are shown in Table 4 and Figure 3. In most of the experimental groups, the cohesive failure of luting materials was the most frequently occurring failure mode (n = 31), followed by mixed type fracture (n = 24 in all groups, especially 30% of the HF and silane-treated group) and adhesive failure between the luting material and dentin (n = 17). Cohesive failure of FRCP showed the lowest incidence (n = 2).

## 4. Discussion

In the present study, the push-out bond strength with the resin cement was higher for FRCPs in the independent silanization groups than for other groups. This result is in accordance with those of previous studies showing a significant increase in bond strength after the silanization of FRCP [21,27], although other studies showed no significant improvement from silanization [18,20]. 

The D.T. Light Post is composed of quartz fibers (60%) and an epoxy resin matrix (40%) [28]. While the quartz fibers have hydroxyl groups to react with a silane coupling agent, the matrix of the D.T. Light Post, the epoxy resin does not have these functional groups [27]. The composition of this matrix leads to controversy regarding the efficacy of silanization, as some previous studies have argued that amino-silane coupling agents do not bond well with an epoxy resin of FRCP [29]. However, in this study, the group with silane and an adhesive bonding agent showed significantly better push-out strength than the group with only adhesive bonding agent, which is explained by silane bonding with the quartz fiber rather than the epoxy resin matrix on the post surface. The bond strength after silanization depends on the substrate. Typically, quartz showed the strongest adhesion forces with resin matrix after silanization because of siloxane (-Si-O-Si-) linkages between the hydroxyl groups on the surface of substrate and silanol (-Si-OH) condensation. This is in contrast to the weak adhesion of –Si-O-M- linkages formed from silanol and pure metals or metal alloys which lack hydroxyl groups [30]. 

Some previous studies have confirmed the benefit of silane by microtensile bond strength testing for translucent fiber posts and dual-polymerizing composite resin core materials [31]. Silane has two functional end groups with different polarities. While the alkoxy group of silanol unit chemically binds with the inorganic silicatized surface, the methacrylate group polymerizes with the composite resin monomers. The reaction between the organic functional groups of silane (with a C=C bond) and functional groups of the resin monomer (with a C=C bond) is induced by the reactive free radicals generated by photo-activation of initiator components in the resin matrix. Reaction of theses free radicals between resin monomers and silane molecules forms a new C-C sigma bond [32]. Since the silane chemically bridges the resin and the OH-covered inorganic substrates [33], it is possible to bond only between the resin luting material and the exposed fibers of FRCP. Some previous studies have explained that sand blasting could remove the surface layer of FRCP, allowing more fibers to react with silane molecules, which can affect the increase in bond strength [21]. However, this study did not include surface alterations of the post by sandblasting treatment according to the manufacturer’s instructions and obtained a better bonding result by silanization alone. Another possible explanation for the beneficial effect of silane application is the improved surface wettability [34]. Once a firm contact is made between the interfacing materials, Van der Waals’ force is effective. This provides a physical bond to activate chemical reactions. In SEM images after post surface treatment, we found that the bonding material was well penetrated into and wrapped evenly on the surface of group 2, the silane and dentin bonding-treated group, compared to group 1 with only dentin bonding without silanization. Therefore, this study of the effects on the silanization of the post surface supports previously published investigations showing the enhanced bond strength of luting materials to the post surface [35].

Alternatively, applying only the adhesive bonding agent to the post surface exhibited significantly lower push-out bond strength in this study. In the case of bonding composite resin luting cement to FRCPs of a cross-linked nature, the surface of the post is well polymerized and little, if any, reactivity is left for free radical polymerization bonding; therefore, no actual chemical bonding is taking place without silanization. In a previous study, when the FRCP with a semi-interpenetrating polymer network polymer matrix was bonded with composite resin luting cement, the interdiffusion bonding could take place by the adhesive bonding agent alone [36]. The authors explained that the linear phase in that material, which was polymethylmethacrylate (PMMA), could be dissolved if a suitable adhesive resin was added on the surface of the post. Therefore, they showed that there were no adhesive failures with the FRCPs, suggesting a better interfacial adhesion of cement to FRCPs by the adhesive bonding agent [36]. In our study, we did not know if BisGMA-based adhesive resin was capable of the dissolving epoxy resin matrix on the post surface and whether this is better than no treatment. We only found that adhesive failure between the post and luting material was observed more frequently in the adhesive alone group (15%) than in the other silanization groups (5%). In SEM images of the adhesive dentin bonding group, the bonding material was flaked off and found mostly floating on the surface, unlike the silane combined group. More research is needed with more specimens available. 

The option of combining chemical and micromechanical surface treatments to improve post retention is known to provide the most promising adhesion-enhancing mechanism [37]. In previous studies, the effects of mechanical surface treatment techniques, such as airborne-particle abrasion, on the bond strength between fiber posts and luting agents were evaluated and were found to be more effective in increasing the bond strength than chemical techniques such as etching with hydrofluoric acid [29]. The manufacturer of the D.T. Light-post did not recommend additional mechanical surface treatment such as sandblasting; we investigated the effect of the HF acid treatment instead of sandblasting. Our results showed no significant bonding strength improvement, which supports the previous investigations into the HF effect. Previous research into the effect of HF acid showed that the application of HF acid not only did not improve the shear bond strength between composite resins and fiber post, but rather caused microcracks and longitudinal fractures within the fiber layer and affected the integrity of the post [17,29,38]. In fact, SEM images of the HF -treated group showed the breakage of fibers and breakdown of resin matrix. However, subsequent silanization and adhesive bonding application could help with luting material penetration and adhesion and might compensate for structural deformation resulting in weakening bonding strength. In the failing fracture mode of this study, there was no significant difference between the HF-treated group and the other groups. 

The self-etching ceramic primer is manufactured to combine etching and silane application step, which helps significantly to shorten the process of conditioning glass ceramic restoration in one step. It is known to be applicable to all methacrylate-based composites and all glass ceramics. In this study, its simplified silanization showed a significantly lower mean push-out strength than independent silane application groups. As a mixture of a ceramic conditioner and a silane coupling agent in one liquid, a simplified single step application followed by a rinse-out process ensures mechanical and chemical surface treatment. Even in a single process, the ceramic conditioning components should be completely removed in order for the silane coupling agent to form strong chemical bonds. However, during the actual application process with the self-etching ceramic primer, the stained (green) components remaining between the fibers were not easily washed away. Residual ceramic conditioning components may be the cause of low bond strength. Unlike glass ceramic restorations, which are the main indication of this product, residual conditioning components in this FRCP seem to interfere with subsequent silanization and adhesion processes. Their SEM image showed the most structural breakdown of fibers and resin matrix of post and less attachment to the bonding material (ceramic primer). Our investigation suggests that a simplified ceramic etching primer is not recommended for the FRCP surface treatment. 

The results of the present study showed that the debonding force in the coronal area is significantly higher than that of the apical area, but in terms of the bond strength per area, the apical area is significantly higher than the coronal area. This result for bond strength is inconsistent with the results of previous studies using a light-curing luting cement [15,35,39]. Multilink N, which was used in this study, is a self-curing luting material with light-curing option for the adhesive luting of FRCP. Unlike light curing luting resin cement, self-curing luting cement may not have the inherent difficulties in moisture control and insufficient light activation of the apical portion, which lead to low bond strength in that portion [40]. 

When evaluating the post retention failure pattern as a result of the experimental method involving the artificial aging process, it is necessary to consider the expansion of the luting material by water absorption and thermal cycling. Previous studies showed that the microscopic examination of all debonded post surfaces and the root canal dentin wall did not show visible any dye penetration; but water molecules could still infiltrate into the FRCP and luting material by diffusion [41,42,43]. The increased water temperature and thermal cycling may have enhanced water diffusion, subsequent hygroscopic expansion, and the significant increase of post retention [44,45]. The luting material expansion consequently caused compressive stresses against the post and dentin surface; the volumetric expansion of composite resin cement attained a value of 1.1% in one study [44]. This hygroscopic expansion may compensate for polymerization shrinkage at different C-factors within the post space depending on the location, but is insufficient [44,45]. Over time, water adsorption would have a deleterious effect on the structure and properties of composite resins [44]. When comparing the push-out strength difference and the fracture patterns according to the post position, the result of the volumetric change by artificial aging mentioned above should also be considered. Meanwhile, previous studies regarding the thermal aging of reinforced thermoset matrix composites frequently revealed a physicochemical degradation of the resin matrix and the loss of adhesion in the matrix /fiber interface [46,47]. These changes could have resulted in lower bond strength and a higher frequency of the cohesive and mixed failure of the luting material in all groups, unlike previous studies without artificial aging [26,40]. 

The limitation of this study is that the dimensions used to calculate the push-out bond strength were not derived directly from the specimens, but from the post-drill dimension. The post interfacial area of the dentin may actually be larger due to variations in root canal morphology. Moreover, centering using the insertion guide devices should have maintained uniform cement thickness in all specimens, but did not do so. The application of various FRCPs and luting cements should also be considered. This study also lacks scientific proof and interpretation about surface and interfacial changes after the silane process. More research is required on surface chemistry properties to determine the relationship between surface treatment of FRCP and bond strength after cementation.

## 5. Conclusions

Based on the findings of this study, the following conclusions were drawn:The silanization of the surface of FRCP before cementation with resin luting cement significantly improved push-out bond strength.Mechanical treatment with hydrofluoric acid prior to the silanization or application of adhesive bonding agent alone to the FRCP surfaces did not significantly improve ret ntion.The self-etching ceramic primer did not significantly improve push-out bond strength.Cohesive failure of luting material was found most frequently in all groups.A significant difference was found in terms of the mean push-out bond strength between the coronal and apical areas.

## Figures and Tables

**Figure 1 materials-12-01585-f001:**
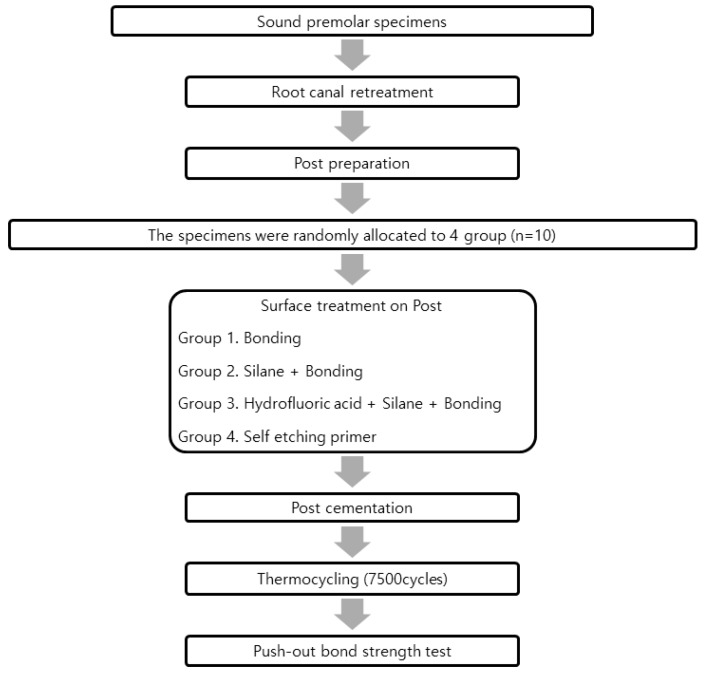
A schematic flow chart of specimen treatment.

**Figure 2 materials-12-01585-f002:**
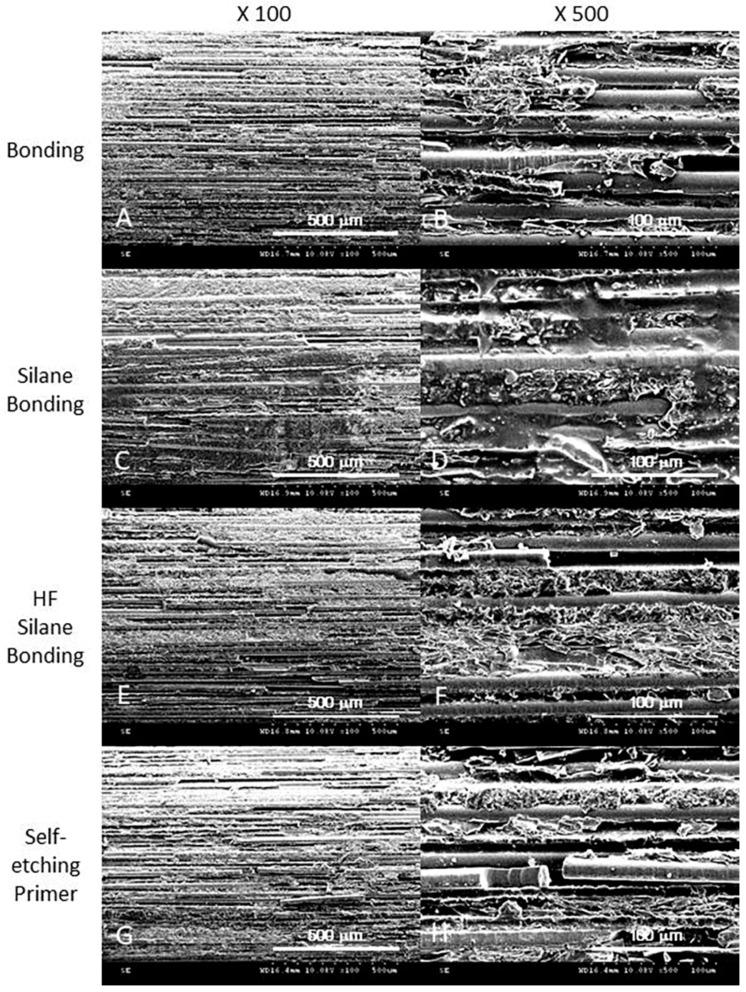
SEM image of the surface of all groups: A and B, dentin bonding group 1; C and D, group 2, silane + dentin bonding treated group; E and F, group 3, HF + silane + dentin bonding treated group; G and H, group 4, self-etching primer treated group.

**Figure 3 materials-12-01585-f003:**
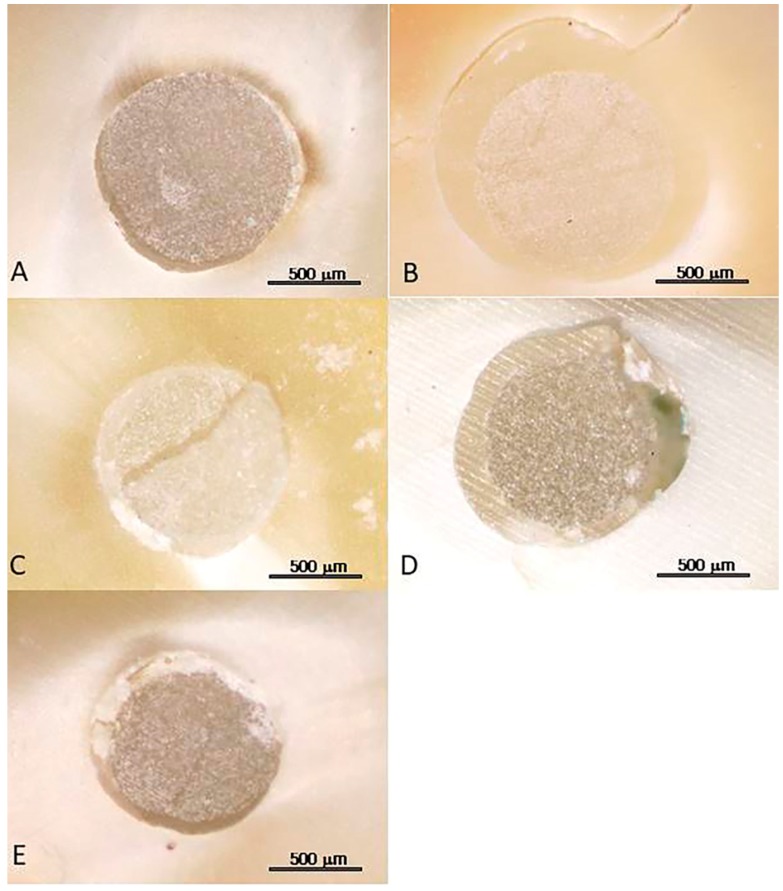
Microscopic views of failed specimens: (**A**) adhesive failure between the luting material and post; (**B**) adhesive failure between luting material and dentin; (**C**) cohesive failure of FRCP; (**D**) cohesive failure of luting material; (**E**) mixed type fracture.

**Table 1 materials-12-01585-t001:** Results of a two-way ANOVA with post surface treatment and post location as the independent variables and debonding force as the dependent variable.

Source	Sum of Squares	df	Mean Square	F	Sig
Corrected model	223.012 ^a^	7	31.859	3.466	0.003
Intercept	26844.479	1	26844.479	2920.553	0.000
Post surface treatment	146.003	3	48.668	5.295	0.002 *
Post location	43.086	1	43.086	4.688	0.034 *
Treatment * location	33.923	3	11.308	1.230	0.305
Error	661.793	72	9.192		
Total	27729.284	80			
Corrected total	884.805	79			

a. R Squared = 0.252 (Adjusted R Squared = 0.179).

**Table 2 materials-12-01585-t002:** Results of a two-way ANOVA with post surface treatment and post location as the independent variables and push-out strength as the dependent variable.

Source	Sum of Squares	df	Mean Square	F	Sig
Corrected model	27.985 ^a^	7	3.998	29.206	0.000
Intercept	413.890	1	413.890	3023.646	0.000
Post surface treatment	1.967	3	0.656	4.790	0.004 *
Post location	25.673	1	25.673	187.552	0.000 *
Treatment * location	0.345	3	0.115	0.840	0.476
Error	9.856	72	0.137		
Total	451.731	80			
Corrected total	37.841	79			

a. R Squared = 0.740 (Adjusted R Squared = 0.714).

**Table 3 materials-12-01585-t003:** Mean value and standard deviation (SD) of debonding force and push-out strength.

Variable		Debonding Force (N)		Push-out Strength (MPa)		Significance
Post Surface	Location	Mean	SD	Mean	SD	
Group 1 (DB)	Apical	16.85	±2.19	2.72	±0.35	A
	Coronal	16.14	±3.54	1.44	±0.31	
	Total	16.49	±2.89	2.08	±0.73	
Group 2 (silane+DB)	Apical	18.82	±2.09	3.04	±0.33	B
	Coronal	20.58	±3.69	1.84	±0.33	
	Total	19.70	±3.06	2.44	±0.69	
Group 3 (HF+silane+DB)	Apical	18.47	±2.68	2.98	±0.43	B
	Coronal	20.58	±2.42	1.84	±0.21	
	Total	19.52	±2.71	2.41	±0.67	
Group 4 (self-etching primer)	Apical	16.19	±3.39	2.61	±0.54	A
	Coronal	18.90	±3.67	1.69	±0.32	
	Total	17.54	±3.71	2.15	±0.64	
Total	Apical	17.58	±2.77	2.84	±0.44	¢
	Coronal	19.05	±3.72	1.70	±0.33	§

**Table 4 materials-12-01585-t004:** Distribution of failure modes according to the experimental groups (%).

Variable		Failure Mode					Total
Post Surface	Location	1	2	3	4	5	
Group 1 (DB)	Apical	2 (20)	2 (20)		4 (40)	2 (20)	10 (100)
	Coronal	1 (10)	2 (20)		5 (50)	2 (20)	10 (100)
	Total	3 (15)	4 (20)		9 (45)	4 (20)	20 (100)
Group 2 (silane+DB)	Apical	1 (10)	2 (20)		4 (40)	3 (30)	10 (100)
	Coronal		2 (20)	1 (10)	5 (50)	2 (20)	10 (100)
	Total	1 (5)	4 (20)	1 (5)	9 (45)	5 (25)	20 (100)
Group 3 (HF+silane+DB)	Apical		4 (40)	1 (10)	3 (30)	2 (20)	10 (100)
	Coronal	1 (10)	2 (20)		2 (20)	5 (50)	10 (100)
	Total	1 (5)	6 (30)	1 (5)	5 (25)	7 (35)	20 (100)
Group 4 (self-etching primer)	Apical	1 (10)	2 (20)		3 (30)	4 (40)	10 (100)
	Coronal		1 (10)		5 (50)	4 (40)	10 (100)
	Total	1 (5)	3 (15)		8 (40)	8 (40)	20 (100)
All group	Apical	4 (10)	10 (25)	1 (2.5)	14 (35)	11(27.5)	40 (100)
	Coronal	2 (5)	7 (17.5)	1 (2.5)	17 (42.5)	13(32.5)	40 (100)
	Total	6 (7.5)	17(21.25)	2 (2.5)	31(38.75)	24 (30)	80 (100)

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
