# Peer review of "Comparison of Self-Etching Ceramic Primer and Conventional Silanization to Bond Strength in Cementation of Fiber Reinforced Composite Post"

_materials, 2019, doi:10.3390/ma12101585_

Reviewer 1 Report

This manuscript represents the study of artificial aging effects on the bond strength of different surface treatments of fibre-reinforced composite posts.The structure of the paper is well organized and the overall paper tells a logical story with a concrete conclusion. It is suitable for publication in Materials after minor revision. Reference section should be improved with published works related to analysis of aging effects on fibre-reinforced composites, as for example:

[1] I. García-Moreno, M.A. Caminero, G.P. Rodríguez, J.J. López-Cela, Effect of thermal ageing on the impact and flexural damage behaviour of carbon fibre-reinforced epoxy laminates, Polymers 11(1) (2019) 80

[2] I. García-Moreno, M.A. Caminero, G.P. Rodríguez, J.J. López-Cela, Effect of thermal ageing on the impact damage resistance and tolerance of carbon-fibre-reinforced epoxy laminates, Polymers 11(1) (2019) 160

Author Response

Revision letter for reviewer 1

Manuscript ID: materials-497341

I would like to express our heartfelt appreciation for your advice. I agree with your comments and have made some corrections in the manuscript. I appreciate your careful and accurate advice and tried to answer it faithfully. The revised part is marked with a blue highlight on the article.

I add the references that you mentioned on the discussion & references. I got the English editing service on MPDI. I hope that our revised manuscript is suitable for publication in Materials.

Yours sincerely,

 Yooseok Shin

Reviewer 2 Report

This article deals with the effect of usage of self-etching ceramic primer on the durability of FRCP. The subject of this work is appropriate for publication in Materials. However, several major revisions described below should be carried out prior to acceptance for publication. Otherwise this manuscript is recommended to be submitted to other journal especially focused on dental materials.

i) First, schientific and technological (i.e. academic) novelty of this work should be stated in the introduction more clearly. In the present form, this work looks like a test of newly launched commercial material rather than an academic research. Moreover, the mechanism of chemical action of the self-etching ceramic primer used in this study should be explained in more detail.

ii) In the discussion, descriptions on the new findings directly derived from this work look vague; in the current form, presentation of the previous studies is the main components of the discussion.

iii) The failure modes have been classified into five criteria on the basis of microscopic observation results. However, the optical microscopy images of the samples have not been given in the manuscript, so that the criterion for distinguishing the failure modes cannot be known. Example of the oprical microscopic images corresponding to the failure modes should be provided.

Author Response

Revision letter for reviewer 2

 Manuscript ID: materials-497341

I would like to express our heartfelt appreciation for your advice.

I agree with your comments and have made some corrections in the manuscript. I hope that our revised manuscript is suitable for publication in Materials. I appreciate your careful and accurate advice and tried to answer it faithfully. The revised part is marked with a yellow and a light green highlight.

I got the English editing service on MPDI. I hope that our revised manuscript is suitable for publication in Materials.

  Yours sincerely,

  Yooseok Shin

Reviewer 3 Report

The manuscript provides a comparative investigation of strength and cyclic ageing (durability) of self-etching ceramic primer in comparison with HF etched and silanization of ceramic phase. The manuscript illustrating comparative data summarized in four tables, appears to my view completely unbalanced, lacking in the discussion of the related questions about morphology, phase texture and surface properties (for example: it is important to show a comparative investigation by electron microscopy identifying morphology and phase(s), resulting etched surface of the quartz fiber in comparison with the native ones, surface chemistry properties as obtained by IR spectroscopy to evidence the efficiency of the silanization and /or etching step). Basing on the before discussed comment, together with standards what I find in majority of the papers published in Materials, I would recommend rejection of the paper.

Author Response

Revision letter for reviewer 3

Manuscript ID: materials-497341

Title: Comparison of Self-Etching Ceramic Primer and Conventional Silanization to Bond Strength in Cementation of Fiber Reinforced Composite Post

I would like to express our heartfelt appreciation for your advice. I agree with your comments and have made some corrections in the manuscript. I hope that our revised manuscript is suitable for publication in Materials.

I appreciate your careful and accurate advice and tried to answer it faithfully. The revised part is marked with a green highlight.

I added SEM images and microscopic images of the surface changes according to the surface treatment method of the post and failure modes. Although not sufficient, we have described the analyzes in the discussion section. Please give a good review.

I got the English editing service from MPDI.

Yours sincerely,

 Yooseok Shin

Round  2

Reviewer 2 Report

The manuscript has been satisfactorily improved. The matters pointed out in the first review were reflected in the manuscript. 

Tips for further improvement:

Scale bars are not clearly seen in the Figure 2 (SEM images). They are required to show the actual size. Similarly, it is better to indicate size of the samples in Figure 3, if possible.

Author Response

Revision letter for reviewer 2

Manuscript ID: materials-497341

Title: Comparison of Self-Etching Ceramic Primer and Conventional Silanization to Bond Strength in Cementation of Fiber Reinforced Composite Post

We would like to express our deep appreciation for your advice.

We tried to answer it faithfully.

The manuscript has been satisfactorily improved. The   matters pointed out in the first review were reflected in the manuscript.

 â˜ž   Thank   you very much. We think that the improvement was done by your careful advices

Tips for further improvement:

Scale bars are not clearly seen in the Figure 2 (SEM images). They are   required to show the actual size. Similarly, it is better to indicate size of   the samples in Figure 3, if possible.

☞ Thank   you for your advice. We add the scale bars in the Figure 2, 3 like follows.

Yours sincerely,

Yooseok

Reviewer 3 Report

I strongly appreciated the experimental work by the authors to implement the previous manuscript, by adding some of the experimental results, before almost all missing. I think that some of the questions have been properly raised (i.e. material structure, possible failures), but no attention has been directed towards the deeper description of the surface chemistry e.g. due to the silanization step. Basing on all these comments, I would summarize that, I think the manuscript is approaching the minimum level of quality for publication in Materials. However, I would encourage the authors to make further study and further implement the manuscript about the surface of samples for the sake of completeness. The aim is manifold: more comprehensiveness and appreciation by readers, more results for the authors and the Journal in terms of citations, a higher contribution to the field.

Author Response

Revision letter for reviewer 3

Manuscript ID: materials-497341

Title: Comparison of Self-Etching Ceramic Primer and Conventional Silanization to Bond Strength in Cementation of Fiber Reinforced Composite Post

We would like to express our deep appreciation for your advices.

I strongly appreciated the experimental   work by the authors to implement the previous manuscript, by adding some of   the experimental results, before almost all missing. I think that some of the   questions have been properly raised (i.e. material structure, possible   failures), but no attention has been directed towards the deeper description   of the surface chemistry e.g. due to the silanization step. Basing on all   these comments, I would summarize that, I think the manuscript is approaching   the minimum level of quality for publication in Materials. However, I would   encourage the authors to make further study and further implement the   manuscript about the surface of samples for the sake of completeness. The aim   is manifold: more comprehensiveness and appreciation by readers, more results   for the authors and the Journal in terms of citations, a higher contribution   to the field.

☞ Thank you very much. We think that the improvement was done by your careful advices

☞ We agree with your points. We admit that we do not have enough knowledge in basic surface chemistry. However, the discussion was added with the best possible knowledge for more comprehensiveness by readers. Please consider it. The revised part is marked with a green highlight.

 4. Discussion

In the present study, the push-out bond strength with the resin cement was higher for FRCPs in the independent silanization groups than for other groups. This result is accordance with those of previous studies showing a significant increase in bond strength after the silanization of FRCP [21,27], although other studies showed no significant improvement from silanization [18,20].

The D.T. Light Post is composed of quartz fibers (60%) and an epoxy resin matrix (40%) [28]. While the quartz fibers have hydroxyl groups to react with a silane coupling agent, the matrix of the D.T. Light Post, the epoxy resin does not have these functional groups [27]. The composition of this matrix leads to controversy regarding efficacy of silanization because some previous studies that amino-silane coupling agents do not bond well with an epoxy resin of FRCP [29]. However, in this study, the group with silane and an adhesive bonding agent showed significantly better push-out strength than the group with only adhesive bonding agent, which is explained by silane bonding with the quartz fiber rather than the epoxy resin matrix on the post surface. The bond strength after silanization depends on the substrate. Typically, quartz showed the strongest adhesion forces with resin matrix after silanization because of siloxane (-Si-O-Si-) linkages between the hydroxyl groups on the surface of substrate and silanol (-Si-OH) condensation. This is in contrast to the weak adhesion of –Si-O-M- linkages formed from silanol and pure metals or metal alloys which lack hydroxyl groups [30].

Some previous studies have confirmed the benefit of silane by microtensile bond strength testing for translucent fiber posts and dual-polymerizing composite resin core materials [31]. Silane has two functional end groups with different polarities. While the alkoxy group of silanol unit chemically binds with the inorganic silicatized surface, the methacrylate group polymerizes with the composite resin monomers. The reaction between the organic functional groups of silane (with a C=C bond) and functional groups of the resin monomer (with a C=C bond) is induced by the reactive free radicals generated by photo-activation of initiator components in the resin matrix. Reaction of theses free radicals between resin monomers and silane molecules forms a new C-C sigma bond [32].

 â˜ž We add the scale bars in the Figure 2, 3 like follows.

Yours sincerely,

 Yooseok Shin
